# Qualitative study: patients' enduring concerns about discussing internet use in general practice consultations

Anita Cuteanu [1], Maureen Seguin,[2] Sue Ziebland,[3] Catherine Pope,[3] Geraldine Leydon,[4] Rebecca Barnes,[5] Elizabeth Murray [1], Helen Atherton,[6] Fiona Stevenson[1]

For numbered affiliations see end of article.

**Correspondence to**
Dr Fiona Stevenson;
f.stevenson@ucl.ac.uk

## ABSTRACT

**Objectives** To examine patients' accounts of their use of the internet before seeing a general practitioner (GP) using thematic analysis of semistructured interviews.

**Design** Qualitative semistructured interview study with transcripts analysed thematically.

**Setting** Primary care patients consulting with 10 GPs working at 7 GP practices of varying sizes and at a range of locations around London and the Southeast of England.

**Participants** 28 adult patients: 16 women and 12 men ranging in age from 18 to 75 from a range of self-defined ethnic backgrounds. Participants were selected based on instances when the patients reported having used the internet before the consultation, when patients referred to the internet in the consultation or when the physician used the internet or made reference to it during the consultation.

**Results** Patients report that they can find health information online that they believe is reliable and helpful for both themselves and their GP. However, they report uncertainty about how to share internet-based findings and reluctance to disclose their efforts at researching health issues online for fear of appearing disrespectful or interfering with the flow of the consultation.

**Conclusions** Despite the democratisation of access to information about health due via the internet, patients continue to experience their use of the internet for health information as a sensitive and potentially problematic topic. The onus may well be on GPs to raise the likelihood (without judgement) that patients will have looked things up before consulting and invite them to talk about what they found.

## INTRODUCTION

The internet, the 'global computer network'[1] connecting and providing us with unlimited information at our fingertips, has become increasingly present in everyday life, with daily use having more than doubled in the last 12 years in the UK.[2 3] It is not surprising that its use for health-related information has followed the same trend,[2] especially as British health authorities and successive governments have increasingly encouraged the population to become more proactive and

### Strengths and limitations of this study

► This study serves to strengthen findings in relation to patients' enduring concerns about discussing their internet use in general practice consultations.

► Only patients who agreed to have their consultation video recorded for the main study ('Harnessing resources from the internet to maximise outcomes from GP consultations') were invited to take part in the interview study.

► Of those, only patients for whom the internet was raised in either their preconsultation survey, or their consultation, were approached.

independent regarding their own healthcare and health management.[3–6] In summary, the internet has become an important source of a wide range of medical information, which is easily accessible by those without medical training.[2 7] Indeed, Professor Helen Stokes-Lampard, former Chair of the Royal College of General Practitioners, highlighted the increasing pervasiveness of the internet in consultations, dubbing its presence as that of 'Dr. Google'.[8]

However, while patients are being encouraged to take on more responsibility and play a greater role in their own healthcare, both general practitioners (GPs) and patients are concerned about how discussions of internet-based health information affect relationships in the consultation.[9 10] Patients report a desire to maximise the available time in consultations by preparing in advance and informing themselves about their problems and possible solutions.[9] Nevertheless, they remain mindful of not wanting to appear to challenge their GP's expertise.[9] Conversely, GPs express anxiety about the potential for their status and expertise to be called into question.[10] In some cases, patients may even find themselves trapped, to at least some degree, in what Bloor and Hororbin[11] described as a 'double-bind',

which expects them to be sufficiently informed, but still defer to their GP's medical expertise and feign ignorance in relation to their medical problem.

This paper offers an in-depth examination of patients' accounts concerning their use of the internet before, during and following GP consultations. Given the aforementioned persistent unease surrounding the use of the internet in the consultation room, we reflect on the ways in which patients discuss and account for their use of the internet with regard to general practice consultations.

## METHODS
### Design
Semistructured qualitative interviews were conducted with 28 patients as part of the Harnessing Resources from the Internet study.[12] The complete data set comprises 281 video-recorded GP consultations, with preconsultation questionnaires completed by all patients, interviews with all 10 participating doctors and 28 selected patients. The 10 GPs came from seven practices across London and the southeast of England. Practices varied in terms of size, population density, level of deprivation and whether they were a training practice. The interviews sought to capture patients' accounts of their use of the internet prior to their video-recorded consultation and reflect on how and why they might, or might not, share prior use of the internet in consultations with their GP.

### Data
We focus on the 28 patients who were interviewed after the consultation. Data were collected between September 2017 and May 2018.

### Data collection
For the initial data collection, up to five patients who had previously indicated they would be prepared to be interviewed were purposively selected (on the basis of survey and consultation data) from each practice and approached. Patients were selected to reflect: (1) instances in which patients reported searching the internet before their GP consultation and this was raised in the consultation, (2) instances in which patients reported searching the internet before their GP consultation and this was *not* raised in the consultation, (3) instances in which the patient raised the topic of the internet in the consultation and this was not reported in the preconsultation questionnaire and (4) instances in which the GP used the internet or raised it as a topic during the consultation.

We sought to provide maximum variation in terms of sociodemographic characteristics. All participants provided signed informed consent which included permission to publish anonymised extracts for the purpose of education and teaching (see online supplemental material 1). To minimise any risk of coercion, it was made clear that the decision to participate or to decline would not affect clinical care and that it was possible to withdraw consent up to the time all data were collected at their site.

We note that in one of the practices, practice number five, no patients agreed to be interviewed. GPs were not told which patients had been approached for a subsequent interview.

Semistructured interviews were conducted by one of two experienced female, non-clinical, qualitative researchers. They took place either in patients' homes or, if preferred, in a room provided by their GP practice. Interviews varied in length from 25 to 90 min and were audio recorded. All were based on an agreed topic guide (see online supplemental material 2), which focused on accounts of what patients did prior to consulting, with prompts relating to the internet if it was not raised. Patients were also asked what happened in the consultation. The interviews were transcribed verbatim and anonymised by a professional transcription service with the transcripts checked for accuracy by the team.

### Data analysis
Data were analysed using thematic analysis.[13–15] Data analysis began with the process of familiarisation, noting down general impressions and providing a content summary for each interview. Initial codes and impressions were carefully sorted and collated into overarching themes.[16] Employing, as Nowell *et al*[16] put it, 'an iterative and reflective process that develops over time and involves a constant moving back and forward between phases' (p. 4). A series of coherent themes were developed, reviewed and defined over a period of several weeks. A data clinic session was held with seven experienced qualitative researchers to help further refine and strengthen the existing themes. Additional discussions and brainstorming sessions with the project leader (FAS) helped structure the themes and develop the narrative.

### Findings
#### Participant characteristics
Interviewees comprised 16 female and 12 male patients ranging in age from 18 to 75 years of age (see table 1).

### Internet research: a complicated but necessary first step for patients?
Although the literature suggests that patients are worried about being perceived negatively, in this study, most patients presented consulting the internet as an important and necessary step prior to seeing their GP.[2] Researching symptoms or concerns on the internet via the well-known web search engine 'Google' has become a reflex for many. This is a testament to its perceived usefulness. The term 'Google' is often automatically cited and creates a feeling of complicity as though its use, in this context, has become part of a now widely shared private joke:

> I: 'Um, okay, and before your appointment with [Dr's name] can you remember whether you sought any sort of information or advice about your symptoms or?'
>
> Pt: 'Google.' (interviewer laughs)

**Table 1** Characteristics of patients interviewed

| Gender | Age (years) | Condition |
|--------|-------------|-----------|
| M | 56–65 | Foot problem. |
| F | 56–65 | Urinary tract infection. |
| M | 56–65 | Suspected inner ear benign tumour. |
| F | 56–65 | Blood in stools. |
| F | 19–25 | Throat feel bruised. |
| F | 56–65 | Blood test result. |
| M | 46–55 | Knee pain. |
| F | 46–55 | Shoulder pain. |
| M | 56–65 | Suspected hernia. |
| F | 66–75 | Requested knee replacement. |
| M | 66–75 | Hole where hernia operation scar is. |
| M | 66–75 | PSA test result. |
| F | 36–55 | Thyroid problem. |
| F | 66–75 | Hissing sound in one ear. |
| F | 26–35 | Pain in ear. |
| F | 26–35 | Complications from deviated septum surgery. |
| M | 0–18 | Baby has oozing belly button. |
| F | 66–75 | Respiratory infection. |
| F | 66–75 | UTI and heart issues. |
| F | 56–65 | Pain behind knee. |
| M | 0–18 | Three-year-old boy has spot on neck. |
| M | 46–55 | Discussion of patient's recovery from throat cancer. |
| F | 56–65 | Review of patient's overactive thyroid. |
| M | 46–55 | Lethargy and pain in chest. |
| M | 56–65 | Check-up for diabetes. |
| M | 66–75 | Review of pain in head. |
| F | 46–55 | Lower back pain, numbness in tip of middle finger. |
| F | 46–55 | Mark on chest. |

PSA, prostate-specific antigen; UTI, urinary tract infection.

I: 'Yeah.'

Pt: Google. Yes, definitely Google. (Pat. 23)

Beyond highlighting that the internet constitutes an important step, patients also report being aware of a need to navigate the information it contains quite carefully. When asked why they prefaced their question to the GP with a disclaimer about having used the internet, a patient explained that although they understood that unfiltered medical information can lead to unnecessary panic, it has become an almost instinctive step:

I repeat it, I know I shouldn't. Yeah 'cause as I said, you know, you tend to be a bit pessimistic and, uh, look at, um, whatever could be the worst thing linked to the, I don't know why, but for some reason, you know, you always look at, I think, the first thing is the, is that you look at it as a sort of first self-help, and, uh,

there's been times where, maybe, I had something and I looked it up. And I thought, 'Oh, phew, it's not that,' as well, you know. Yeah. Erm, obviously cancer is the number one, um, and all sorts of things, but, um, yeah, I think, sometimes, internet can be useful. (Pat. 28)

Linked to this is that although looking on the internet can be helpful, there is a need to remain vigilant in terms of assessing the information, with a conclusion that consulting is the best solution to medical concerns.

I think, sometimes, internet can be useful. Erm, maybe, you know, if you have a little, um, problem, or you can look it up and you can have a better idea of what it is. Er, but again, you know, sometimes it can be on, on the other hand it could be quite misleading, because then you'll panic. So, it's two ways, you have to sort of know how to, probably how to use it and what to do with it. And always better to see a doctor, obviously. (Pat. 28)

Some people reported using the internet to solve their medical problems, and only when they failed to solve their problem 'resorted' to consulting a doctor.

So, I did try and find out as much information as I could, but, erm, I wasn't very successful so, that's why. […] I resorted to going to the doctor. […] That's the only thing I did do, I just went on the internet, that's all. (Pat. 10)

Using the internet was described as serving a preparatory function in ensuring patients were equipped with information to facilitate their upcoming consultation. The internet was seen to present the offer of an advantage in terms of knowing the right questions to ask, enabling patients to navigate towards an optimal management or treatment outcome.

You can ask the doctor more about what you've got because you've got more information about what you've got. I'm not saying that you distrust doctors or anything like that. But in this day and age with everybody under pressure, it's best to know what you're coming for. […] So it, you know, you wanna be able, you wanna have knowledge 'cause you wanna be able to not get fobbed off with anything less than getting [the issue] fixed. […]

And in fact, they do say that people who are more eloquent with, with what they, how they can describe […] they get the better treatment. […] The doctors are highly intelligent, so if you go in there and […] have a good conversation, you're armed with information, you can ask the right questions and lead to the right treatment and try and get that treatment. (Pat. 6)

When asked what benefits they perceived from using the internet, some patients felt that the internet helped them to optimise limited time in the consultation room:

You know, you've got limited time to get things across. (Pat. 3)

Finally, it is important to note that patients generally reported the need to reflect on their use of the internet. For example, in relation to when information available online would stop being useful and become instead a source of anxiety:

But you really need to have, I think, a very steady mind (patient laughs) um, to use that. So, I kind of, um, rather than not looking on internet, I would say I look on internet, but keeping a sort of, um, protecting myself, you know, from going too far. (Pat. 28)

### Use of the internet as a reflection of the doctor–patient dynamic

In line with existing literature,[2 9 10] concern was expressed that it was not appropriate to mention use of the internet prior to consulting. This was associated with a perceived procedural etiquette when talking to their doctor.

I: 'In the consultation I don't think you sort of referred to the fact that you'd gone on the internet, was there any reason why you didn't sort of bring that up or?'

Pt: 'Um, I think I always think, doctors probably don't want to hear that, because you, you've gone to ask them.'

I: 'Mhm.'

Pt: 'Haven't you? So, um, I don't really want to say, 'Well, I've looked it up on the internet what do you think?' (Pat. 23)

Patients talked of the need to monitor for the right slot to mention searching as well as a concern to ensure that mentioning use of the internet is not perceived as an explicit request or pressure for a particular GP action, such as prescribing.

As I say, it's one of those things you sort of make known at the appropriate point if one arises. And not sort of, go in with a banner over your head saying, 'I've looked all this up on the internet. Now just give me a prescription brother'. Not the sensible way of doing it. (Pat. 3)

Some patients reported that their understanding of how interactions between doctors and patients worked meant they struggled to see how they could integrate what they learnt from the internet into the consultation.

But on the whole, I don't think it's the sort of thing the GP wants to discuss. […] It doesn't fit my model, or what I perceive to be the GP's model, of how the conversation is supposed to go. (Pat. 11)

Interestingly the apparent sensitivity to discussing use of the internet for health was also evident in the patients' responses to the researcher, in which their internet use was presented as prudent, rational or 'common sense',

and in some cases, patients emphasised legitimising personal characteristics such as a professional background, for example, healthcare or education, that worked to normalise their use of the internet

So, I mean, because I was a teacher, just Googling and looking for information and sifting out information, something I did all the time, […] as a (subject) teacher that you have to do that. (Pat. 6)

## DISCUSSION
### Summary
Patients reported use of the internet as an initial response to health symptoms and a way of reducing the pressure on the UK healthcare system. Their reports of soliciting medical expertise only after seeking information online is consistent with previous research.[2] It is however notable that despite the internet being presented as a key way in which patients managed their health, caution was expressed by patients about both use and sharing internet use with the GP, and in some cases, use of the internet was presented as a deviant action.

### Strengths and limitations
This analysis extends previous research about patients' use of, and views about, sharing information from the internet in primary care consultations. Only patients who agreed to have their consultation video recorded for the main study and indicated they were happy to be contacted for an interview were invited to take part in the interview study. Of those, only patients for whom the internet was raised in either their preconsultation survey, or their consultation, were approached. This means our sample focused on those people for whom the internet had played a part in a recent consultation grounding data collection in an actual event as opposed to hypothetical ideas.

### Comparison with existing literature
The findings are consistent with the existing literature regarding patients' reluctance to share details of their internet research despite internet use continuing to intensify over the years.[17 18]

Our findings are in line with Parsons' now classic sociological theory relating to medical interactions first outlined nearly 70 years ago.[19] Parsons pointed to the functional asymmetry of the medical encounter and how the medical consultation is itself embedded within the wider functionality of the institution of medicine in society.[19] While this may on the face of it appear undesirable and a barrier to open communication between the doctor and their patient, detailed analysis of medical interactions in a range of settings have illustrated that asymmetry is key to understanding the ways in which both medical professionals and their patients organise their interactions.[20] One way to engage with and position our findings is in relation to the idea discussed by Heritage[21] of access and

rights to specific domains of medical knowledge and information in medical consultations. The avoidance by patients of broaching their use of the internet explicitly or directly with their healthcare providers can be seen as in keeping with attempts by patients to maintain the shared understandings in relation to how medical consultation operate and not be seen to encroach, uninvited, on the doctor's 'domain' of medical knowledge.

## Implications for research and/or practice

These observations highlight a difficulty in balancing the perceived responsibility of being a patient in the internet age on the one hand with the presentation of a feeling of needing to conform to a prior model in which 'medical knowledge' is only accessible via the doctor. There is an opportunity for GPs to take the initiative when patients present their problem at the beginning of the consultation to ask what they think might be wrong and if they have looked the problem up.

## CONCLUSION

Previous research on this topic showed that there was a reluctance on the part of patients regarding sharing information found on the internet. Our findings support this claim and show that despite an ever-increasing democratisation of access to information and growth in use of the internet across the population, this reluctance is still present. This study offers insights into how patients approaching use of the internet for health concerns. Such understandings are crucial given the current pandemic is and will continue to encourage a greater adoption of, reliance on, technology and the internet in particular.

**Author affiliations**
[1]Research Department of Primary Care and Population Health, University College London, London, UK
[2]Department of Health Services Research and Policy, London School of Hygiene & Tropical Medicine, London, UK
[3]Nuffield Department of Primary Care Health Sciences, University of Oxford, Oxford, UK
[4]Primary Care and Population Sciences, Faculty of Medicine, University of Southampton, Southampton, UK
[5]Centre for Academic Primary Care, School for Social and Community Medicine, Bristol University, Bristol, UK
[6]Unit of Academic Primary Care, Warwick Medical School, University of Warwick, Coventry, UK

**Acknowledgements** The authors would like to thank the participants of this study as well as the Harnessing Resources from the Internet project research team for their support and collaboration. We would particularly like to acknowledge the work of Laura Hall who was one of the researchers on the study.

**Contributors** FAS, GL, RB, SZ, CJP, EM and HA conceived of the study, designed it and securing funding. MS collected the data. Analysis of the data and initial drafts were led by AC and FA and the manuscript was reviewed critically by GL, RB, SZ, CJP, EM, HA and MS. All authors have edited and approved the manuscript for submission.

**Funding** The HaRI project was funded by the National Institute for Health Research School for Primary Care Research (Award/Grant Number 284). National Health Service (NHS) costs are covered via the Local Clinical Research Network. The present study has received funding from the Laidlaw Foundation.

**Disclaimer** The views expressed are those of the authors and not necessarily those of theNational Institute for Health Research, the NHS or the Department of Health.

**Competing interests** None declared.

**Patient and public involvement statement** Two study participants and public involvement representatives advised the research group from the inception of the study. They were also members of the steering committee and advised throughout the study.

**Patient consent for publication** Not required.

**Ethics approval** The HaRI study received ethics approval from the UK NHS Research Ethics Committee and governance approval obtained from the Health Research Authority. All participants provided written consent before taking part in the study.

**Provenance and peer review** Not commissioned; externally peer reviewed.

**Data availability statement** No data are available. Some data are available for reuse on application following appropriate ethics and data governance requirements.

**ORCID iDs**
Anita Cuteanu http://orcid.org/0000-0001-9490-5991
Elizabeth Murray http://orcid.org/0000-0002-8932-3695

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
