## [Reviewer comments · BMJ Open]

ARTICLE DETAILS

TITLE (PROVISIONAL)	Qualitative Study: Patients' Enduring Concerns about Discussing Internet Use in General Practice Consultations
AUTHORS	Cuteanu, Anita; Seguin, Maureen; Ziebland, Sue; Pope, Catherine; Leydon, Geraldine; Barnes, Rebecca; Murray, Elizabeth; Atherton, Helen; Stevenson, Fiona

VERSION 1 – REVIEW

REVIEWER	Andrikopoulou, Elisavet University of Portsmouth Faculty of Technology, School of Computing
REVIEW RETURNED	25-Jan-2021

GENERAL COMMENTS	References 3 and 4 lead to a 404 error of page not exists.
--

REVIEWER	Archibald, Douglas University of Ottawa Faculty of Medicine, Family Medicine
REVIEW RETURNED	10-Feb-2021

GENERAL COMMENTS	Overall I found this to be a very interesting paper that will have wide appeal to the readership of BMJ OPEN. The investigators have done an adequate job of ensuring diversity and representation of patients for the interviews. The findings are well described and the discussion and conclusion are justified. Specific comments are as follows: 1. My major concern is that there is no methodology. Interviews are a method but not a methodology on their own. I was pleased to read the authors have included the COREQ checklist. However, item #9 clearly states, "What methodological orientation was stated to underpin the study? e.g. grounded theory, discourse analysis, ethnography, phenomenology, content analysis". The authors have indicated this can be found on p. 5 but I could not find it.2. Major concern. How were patients recruited for this study? All I read is participants were purposively selected from each practice. As a reader I would like to know more. Are there issues with purposive selection? Was coercion an issue?3. Minor concern. I believe internet should be capitalized as it is a proper noun. Perhaps this is a decision for the editor.4. Period is missing after the references on p.6 line 17.
--

	5. I do like the subheadings of the discussion. However, "How this fits in" should really be the "conclusion".
--	--

VERSION 1 – AUTHOR RESPONSE

Reviewer 1:

Comment: "References 3 and 4 lead to a 404 error of page not exists."

Response: Apologies, I have updated the sources with working links.

Reviewer 2:

Comment: "1. My major concern is that there is no methodology. Interviews are a method but not a methodology on their own. I was pleased to read the authors have included the COREQ checklist. However, item #9 clearly states, "What methodological orientation was stated to underpin the study? e.g., grounded theory, discourse analysis, ethnography, phenomenology, content analysis". The authors have indicated this can be found on p. 5 but I could not find it. "

Response: I have amended the manuscript to explicitly reflect the methodological approach both in the abstract and main manuscript following the example of recently published studies in BMJ Open which took a similar methodological approach.

Comment: "2. Major concern. How were patients recruited for this study? All I read is participants were purposively selected from each practice. As a reader I would like to know more. Are there issues with purposive selection? Was coercion an issue?"

Response: The patients were recruited from seven practices where the data collection was taking place for the Harnessing Resources from the Internet (HaRI) study. Patients were asked to indicate on the consent form for the recording of the consultation if they would be happy to be interviewed and provide contact details. These interviews were conducted after the recording of their consultation. This has been specified in the manuscript.

As stated in the manuscript up to five potential interviewees were selected for each practice based on whether (i) they had indicated they had used the internet prior to their consultation, and this fact was raised during their consultation, (ii) they had indicated they had used the internet prior to their consultation, and this fact was not raised during their consultation, (iii) patients referred to the internet during the consultation but did not report prior use in the pre-consultation questionnaire, or (iv) the GP referred to the internet in the consultation independently of the patient's report.

Coercion is unlikely to have been an issue as patients were given a choice to be interviewed independently of participation in the rest of the study, i.e. being interviewed was not a condition of taking part in the main study. As highlighted in the 'Data Collection' sub-heading, it was made clear that the decision to participate or to decline would not affect clinical care and that it was possible to withdraw consent up to the time all data were collected at their site. We can also report that in one instance, for Practice #5, no patients agreed to be interviewed indicating it was unlikely potential interviewee felt coerced to participate. The GPs themselves were not made aware which patients had been approached to minimise any risk of coercion.

Comment: "3. Minor concern. I believe internet should be capitalized as it is a proper noun. Perhaps this is a decision for the editor."

Response: I agree that this is ultimately a decision for the editors.

Comment: "4. Period is missing after the references on p.6 line 17."

Response: Thank you, I have added the correct punctuation.

Comment: "5. I do like the subheadings of the discussion. However, "How this fits in" should really be the "conclusion"."

Response: Thank you for the suggestion. I have amended the heading accordingly.

VERSION 2 – REVIEW

REVIEWER	Archibald, Douglas University of Ottawa Faculty of Medicine, Family Medicine
REVIEW RETURNED	12-Apr-2021
GENERAL COMMENTS	The authors have adequately addressed my concerns.